# Controllable photomechanical bending of metal-organic rotaxane crystals facilitated by regioselective confined-space photodimerization

Jun-shan Geng[1,2], Lei Mei [1✉], Yuan-yuan Liang[1], Li-yong Yuan[1], Ji-pan Yu[1], Kong-qiu Hu[1], Li-hua Yuan [2], Wen Feng [2✉], Zhi-fang Chai[1,3] & Wei-qun Shi [1✉]

Molecular machines based on mechanically-interlocked molecules (MIMs) such as (pseudo) rotaxanes or catenanes are known for their molecular-level dynamics, but promoting macro-mechanical response of these molecular machines or related materials is still challenging. Herein, by employing macrocyclic cucurbit[8]uril (CB[8])-based pseudorotaxane with a pair of styrene-derived photoactive guest molecules as linking structs of uranyl node, we describe a metal-organic rotaxane compound, U-CB[8]-MPyVB, that is capable of delivering controllable macroscopic mechanical responses. Under light irradiation, the ladder-shape structural unit of metal-organic rotaxane chain in U-CB[8]-MPyVB undergoes a regioselective solid-state [2 + 2] photodimerization, and facilitates a photo-triggered single-crystal-to-single-crystal (SCSC) transformation, which even induces macroscopic photomechanical bending of individual rod-like bulk crystals. The fabrication of rotaxane-based crystalline materials with both photoresponsive microscopic and macroscopic dynamic behaviors in solid state can be promising photoactuator devices, and will have implications in emerging fields such as optomechanical microdevices and smart microrobotics.

[1] Laboratory of Nuclear Energy Chemistry, Institute of High Energy Physics, Chinese Academy of Sciences, 100049 Beijing, China. [2] Key Laboratory of Radiation Physics and Technology of the Ministry of Education, Institute of Nuclear Science and Technology, College of Chemistry, Sichuan University, Chengdu 610064, China. [3] Engineering Laboratory of Advanced Energy Materials, Ningbo Institute of Industrial Technology, Chinese Academy of Sciences, Ningbo 315201 Zhejiang, China. ✉email: meil@ihep.ac.cn; wfeng9510@scu.edu.cn; shiwq@ihep.ac.cn

As a mimic of the macroscopic counterparts, artificial molecular machines[1–3] that can undergo precisely controlled dynamic motion at the molecular level upon stimulation by external signals such as light[4,5], electricity[6] or chemical reagents[7] have attracted considerable attention in recent years. Mechanically interlocked molecules (MIMs) including rotaxanes[8,9], catenanes[10,11], and other molecular assemblies with more sophisticated structure[12,13], which incorporate different kinds of organic macrocyclic molecules such as crown ether[14,15], cyclodextrin[16] and cucurbituril[17] and feature intrinsic dynamic nature, have been often utilized as key constituent components of certain molecular machines[18,19]. To date, most molecular machine systems are designed to work in solution where each work unit is isolated from each other and functions independently and incoherently. For example, a variety of molecular machines reported so far including molecular switches[20,21], molecular pump[22], molecular motors[23–25], molecular muscle[26,27], and molecular robot[28] are capable of making increasingly complex operations in aqueous or nonaqueous environments, where different sets of supramolecular motifs are dispersed and separated individually in solution. However, when extended to a solid-state system with significantly shortened intermolecular distances and massive intermolecular interactions that is dramatically different from a solution environment, it is still challenging to construct MIM-based molecular assemblies with structure dynamics in solid. It is supposed that the dynamic performance of MIMs in such a condensed state could be largely inhibited by great steric hindrance of neighboring atoms and a large number of weak interactions between them.

To reduce the influence of surrounding environment on MIM motifs in solid, a feasible method would be to place the dynamic MIM components into solid-state materials with large pores such as metal-organic frameworks (MOFs) with attractive properties, such as easy synthesis, controllable structure, multiple functions and high thermal and chemical stability[29–35]. For example, when rotaxane or pseudorotaxane is utilized as the linker of MOF materials, the MIM unit of rotaxane would have more free space in the resultant metal-organic rotaxane framework (MORF) materials, and could be able to undergo motion of rotation, translation or isomerization inside this kind of porous materials[14,15,36–43]. Nevertheless, the dynamic changes of MIM motifs in those solid-state materials reported so far are merely restricted at the molecular- or nano- scale, whilst the synergistic effects between different MIM units in solid have been rarely exploited[44]. Solid-state molecular machines have, as compared to molecular machines working in solution, a higher degree of molecular organization and structural order due to the characteristics of condensed phases. If the many molecules in solid state, especially in the crystalline state, can be appropriately arranged through reasonable assembly strategies to effectively accumulate molecular-scale strain or stress, it is possible to realize the conversion of microscopic structural changes to macroscopic deformation or motion of the bulk material for these solid-state molecular machines, just like an actuator that can transform external energy to mechanical work (Fig. 1a).

For the design of macroscopic actuator, there are two key issues that need to be taken into account: the external energy input to trigger molecular-level dynamics and structure assembly strategies to realize the conversion from molecular level to macroscopic level. Among different forms of energy stimuli including mechanical force or pressure, heat, light and electric fields, light is one of the most attractive forms since it possesses several excellent attributes such as remote non-destructive control, adjustable wavelength, intensity and polarization, and easy operation[45,46]. Furthermore, the energy-conversion path for these light-controlled actuators (namely photoactuators) relies mostly on photochemical actuation involving reversible shape-changing photoreactions (such as photoisomerization, photodimerizaiton and photocycloaddition) of photoactive groups activated by visible or ultraviolet (UV) light. A typical case of photochemical reaction is the photodimerization of olefin derivatives first reported by Schmidt et al[47] as early as 50 years ago, of which the proceeding in solid requires proper stacking arrangement of photoactive ligands in space and strict distance between the photoactive ligands[48–50]. Different strategies such as using small organic molecules[51], metal organic skeleton[52], coordination with metal cations[53], cation-π interaction[54] and supramolecular encapsulation[55] are employed to guarantee the structure requirement of photodimerization. In MIM-based actuator, this structural requirement could be fulfilled by elegant design of rotaxane units through simultaneously confining two photoactive guests in a macrocyclic cavity. Besides the energy route to achieve molecular dynamics, the structure assembly is also crucial to the macroscopic actuation of actuators, as it can realize the accumulation and amplification of molecular-scale stress caused by molecular deformation. Exactly, the anisotropic assembly and packing of crystalline MORF materials would be very helpful to effectively reduce the stress dissipation. Meanwhile, placing photoactive groups on the material framework, like incorporating the photoactive axles as linking struts into MORFs to promote greater photo-induced structural changes and internal stress could be a feasible architectural design strategy.

Herein, a bifunctional styrene-derived photoactive ligand (E)-4-[2-(methylpyridin-4-yl) vinyl]benzoic acid ([HMPyVB]I) is designed, which can form a [G$_2$@H] psudorotaxane (G: guest molecule, H: host molecule) with a pair of aromatic moieties of [HMPyVB]$^+$ or MPyVB encapsulated in the cavity of cucurbit[8]uril (CB[8]) macrocycle, and further coordinate with metal ions through the deprotonated carboxylate group (Fig. 1b). By utilizing the in-situ assembly between [HMPyVB]I, CB[8] and uranyl ion, we report a kind of photoresponsive MORF, U-CB[8]-MPyVB, that contain photoactive MPyVB guest molecules trapped in the cavity of macrocyclic CB[8] host. Benefitting from the presence of (MPyVB)$_2$@CB[8] linker, the UV light-induced molecular dynamics in the solid-state U-CB[8]-MPyVB sample is achieved via a regioselective [2 + 2] photodimerization-based single-crystal-to-single-crystal (SCSC) transformation, and examined in detail by using a set of characterization techniques including single-crystal X-ray diffraction analysis, $^1$H NMR, IR and fluorescence spectra. More strikingly, the photodimerization reaction at molecular scale even induces macroscopic photomechanical bending of individual bulk crystals, which demonstrates a photoactuation system based on photoresponsive MORFs. The macroscopic deformation of this crystalline photoactuator has been characterized, and a deep understanding of photomechanical actuation mechanism of U-CB[8]-MPyVB has been also provided through a comprehensive analysis and comparison.

## Results

**Synthesis and structure characterization of U-CB[8]-MPyVB.** The synthesis of MORF compound, U-CB[8]-MPyVB, is firstly conducted via a one-pot hydrothermal reaction. Through assembly of UO$_2$(NO$_3$)$_2$·6H$_2$O, [HMPyVB]I and CB[8] under hydrothermal conditions, yellow rodlike crystals of U-CB[8]-MPyVB were obtained. Single crystal X-ray diffraction (SC-XRD) analysis reveals that U-CB[8]-MPyVB belongs to triclinic system with space group P-1, which has three MPyVB ligands, one and a half of CB[8] and a tetranuclear uranyl in the asymmetric unit as illustrated in Fig. 2a. It is noteworthy that although the coordination geometries of the four uranyl ions are both pentagonal

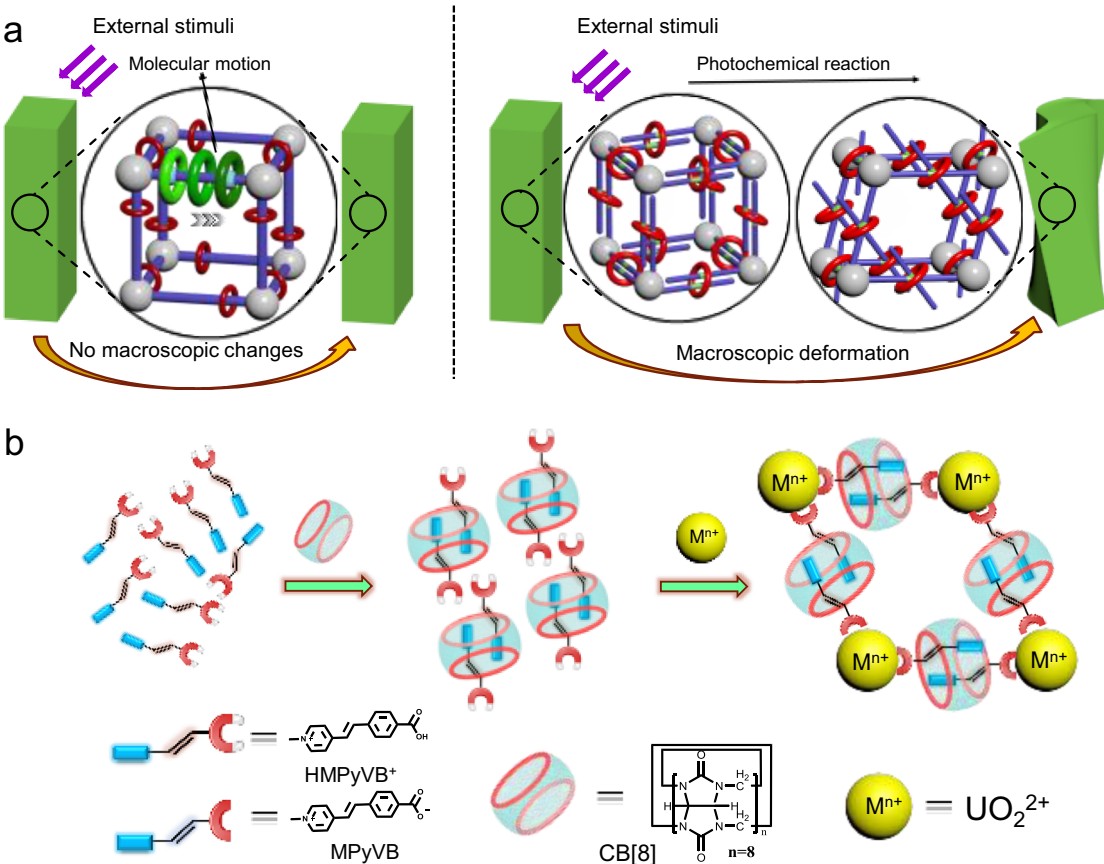

**Fig. 1 Macroscopic and microscopic changes in the dynamic MORFs in solid state by external stimuli and synthetic diagram of U-CB[8]-MPyVB.**
**a** Macroscopic and microscopic changes in the dynamic MORFs in solid state by external stimuli (The figure on the left shows some movement on the micro level but with no changes on the macro level, the figure on the right shows the microstructure and macromorphology changes at the same time). **b** Synthetic diagram of U-CB[8]-MPyVB.

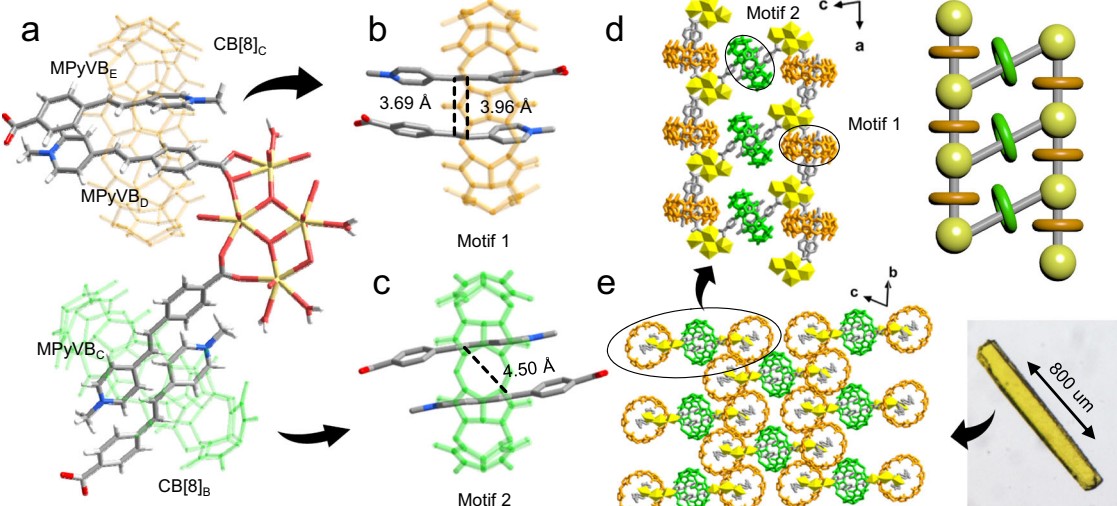

**Fig. 2 Single crystal structure of U-CB[8]-MPyVB.** a Asymmetric unit of U-CB[8]-MPyVB. Enlarged motifs (**b**, Motif 1 with orange-colored CB[8]; **c**, Motif 2 with green-colored CB[8]) showing different distances between C=C bonds in a pair of MPyVB ligands. **d** One-dimensional chain of U-CB[8]-MPyVB connected by a combination of supramolecular inclusion and coordination interactions (Color code: oxygen atom, red; carbon atom, gray; nitrogen atom, blue; uranium atom, yellow). **e** Packing mode of U-CB[8]-MPyVB in 3-D space and morphology of crystal structure (insert: optical photograph of a crystal of U-CB[8]-MPyVB).

bipyramids, their coordination environments are different. A close inspection reveals that MPyVB ligands and CB[8] macrocycles, which are denoted as $MPyVB_C$, $MPyVB_D$ and $MPyVB_E$, $CB[8]_B$ and $CB[8]_C$ for the ease of discussion, can be sorted into two $(MPyVB)_2@CB[8]$ moieties with different assembly modes (Supplementary Table 1 and Supplementary Table 2). Two nonparallel $MPyVB_D$ and $MPyVB_E$ are encapsulated by $CB[8]_C$ and the distance of C=C double bonds between them is 3.69 Å and 3.96 Å (Motif 1, Fig. 2b), which meets the Schimdt's topochemical criteria (<4.2 Å) in solid state. By means of supramolecular inclusion and coordination linkage, two infinitely extended 1-D metal organic rotaxane chains are achieved along a-axis. These two one-dimensional chains are connected by Motif 2 $((MPyVB_C)_2@CB[8]_B$, Fig. 2c) to form a ladder-like chain (Fig. 2d). The distance between two C=C double bonds from this pair of $MPyVB_C$ is 4.50 Å, which seems to be too long to meet the requirement of [2 + 2] photodimerization reaction. Furthermore, noncovalent interactions including hydrogen bonding, π···π stacking and C-H···π interactions between adjacent CB[8] moieties finally lead to the formation of a 3-D framework structure (Fig. 2e). Powder X-ray diffraction (PXRD) pattern of U-CB[8]-MPyVB shows that the experimental peaks of the crystals fit well with the simulated ones from single crystal data (Supplementary Fig. 2). Thermogravimetric analysis shows that U-CB[8]-MPyVB has no weight loss until 300 °C, indicating its good thermal stability (Supplementary Fig. 3). In IR spectrum of U-CB[8]-MPyVB, a typical U=O vibration band at 907 cm$^{-1}$ shows the presence of uranyl in this system (Supplementary Fig. 4). Compared to strong fluorescence of supramolecular complex CB[8]-HMPyVB, the fluorescence of U-CB[8]-MPyVB is greatly weakened (the fluorescence quantum yield changes from 2.8% for CB[8]-HMPyVB to ~0% for U-CB[8]-MPyVB after uranyl coordination), accompanied by a slight red shift (Supplementary Fig. 5, emission wavelength is 526 nm). The fluorescence intensity of U-CB[8]-MPyVB is only 12.4% of CB[8]-HMPyVB, which indicates that uranyl ion quenches the fluorescence of CB[8]-HMPyVB through a metal-to-ligand charge transfer process.

An interesting question is how U-CB[8]-MPyVB can be assembled by a one-pot method. Given possible competition between metal coordination of MPyVB and its supramolecular encapsulation by macrocyclic CB[8] during the formation process of U-CB[8]-MPyVB, two possible assembly routes are proposed: (1) Deprotonated MPyVB first coordinates to uranyl to form an intermediate of uranyl-MPyVB complex, which is further connected by CB[8] through supramolecular inclusion to obtain the final MORF (Fig. 3a). A test experiment employing a different feeding order, i.e., firstly mixing $UO_2(NO_3)_2 \cdot 6H_2O$ and [HMPyVB]I followed by the addition of CB[8] macrocycles, shows that, a large amount of precipitate generate immediately after simply mixing $UO_2(NO_3)_2 \cdot 6H_2O$ and [HMPyVB]I in the aqueous solution, which remains unchanged after hydrothermal treatment. Similar phenomenon is observed even if an interface diffusion method is used to reduce the contact velocity between $UO_2(NO_3)_2$ solution and [HMPyVB]I solution (molar ratio is 1:1). The precipitation effect between [HMPyVB]I and uranyl should be attributed to strong coordination bonding between uranyl and the carboxylate group of [MPyVB], which has been identified in several previously-reported uranyl complexes with similar pyridium- or viologen-functionalized organic carboxylate linkers that are insoluble in aqueous solution (Supplementary Fig. 6)[56–58]. Since the emerging of the precipitation reaction between [HMPyVB]I and uranyl will prevent pseudorotaxane formation between [HMPyVB]I and CB[8], the assembly mechanism of coordination followed by supramolecular inclusion can be excluded. (2) [HMPyVB]I ligands first assemble with CB[8] via supramolecular encapsulation to produce the CB[8]-HMPyVB

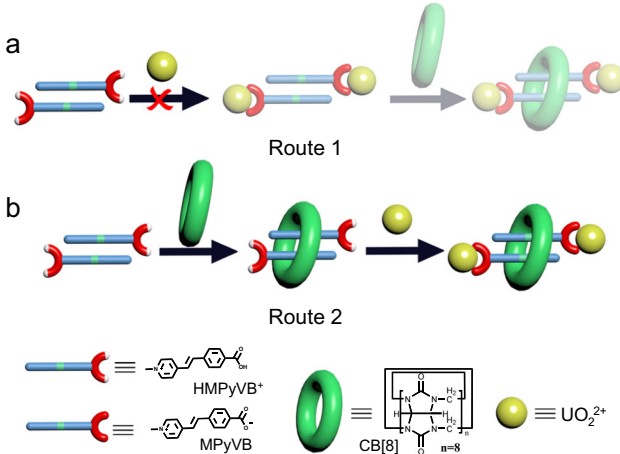

**Fig. 3 The possible formation mechanism of U-CB[8]-MPyVB. a** Route 1, uranyl coordination followed by supramolecular inclusion. **b** Route 2, supramolecular inclusion followed by uranyl coordination.

intermediate, and then further deprotonate and coordinate with uranyl to form the final uranyl-based MORF compound (Fig. 3b). To verify this assumption, $^1H$ NMR and ESI-MS experiment were performed to confirm the existence of stable $(HMPyVB^+)_2@CB[8]$ host-guest complex (Supplementary Fig. 7 and Supplementary Fig. 8). In addition, single crystal of CB[8]-HMPyVB intermediate was also successfully isolated, and single crystal structure analysis shows that two $HMPyVB^+$ motifs are encapsulated by a CB[8] host with the distance between C=C double bonds in them is 4.44 Å (Supplementary Fig. 9). Furthermore, the hydrothermal reaction of deprotonated CB[8]-HMPyVB complex with uranyl gives the final product U-CB[8]-MPyVB. These above experiments strongly support that route 2 should be the possible way to the formation of this elegant MORF compound.

**Regioselective solid-state [2 + 2] photodimerization in light irradiated U-CB[8]-MPyVB crystals.** Because of the existence of Motif 1 among which the distance between two C=C double bonds meets the Schimdt's topochemical criteria, we are curious whether U-CB[8]-MPyVB is photoresponsive under light excitation. Therefore, a rod-shaped crystal of U-CB[8]-MPyVB immersed in mineral oil was selected for light irradiation testing. Interestingly, after irradiation by UV light for a period of time, this crystal starts to move around in the mineral oil, accompanied by slight macroscopic bending (Supplementary Fig. 10 and Supplementary Movie 1). A control experiment by directly heating crystals in mineral oil over the temperature range from room temperature (~16 °C) to the maximum temperature induced by photothermal effect (~30 °C) reveals that pristine crystals of U-CB[8]-MPyVB keep silent and show no sign of motion (Supplementary Fig. S11), thus excluding the possibility of crystal motion caused by thermal disturbance. This interesting photoresponsive phenomenon indicates that the photo-triggered reactions may occur within the crystal of U-CB[8]-MPyVB during incident irradiation. The $^1H$ NMR experiment reveals that there is photodimerization product, 1,3-bis (4-methylpyridin)-2,4-bis (benzoic acid) cyclobutane (bisMPyVB), in the digested sample of U-CB[8]-MPyVB. Specifically, U-CB[8]-MPyVB-A was digested with 20 μL concentrated hydrochloric acid (37%) and diluted by DMSO-$d_6$ (0.5 mL), then subjected shortly to $^1H$ NMR spectroscopy. As is shown in Fig. 4a, the integration of the MPyVB part (existing in the form of [HMPyVB]$^+$ in hydrochloric acid

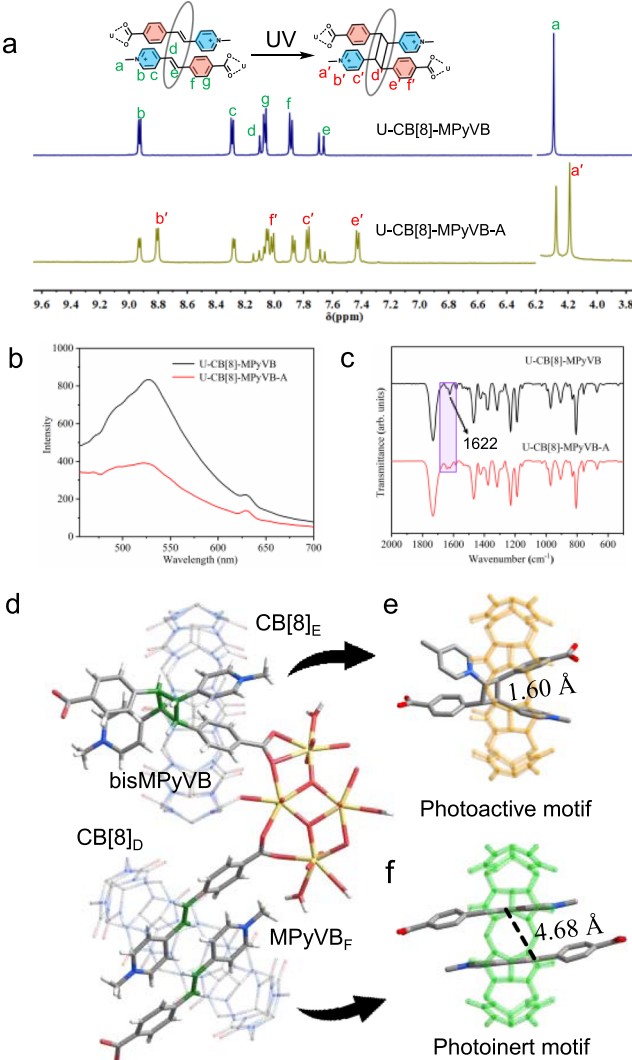

**Fig. 4 A comparison of U-CB[8]-MPyVB and U-CB[8]-MPyVB-A. a** A comparison between solid fluorescence spectra of U-CB[8]-MPyVB and U-CB[8]-MPyVB-A. **b** A comparison between FTIR spectra of U-CB[8]-MPyVB and U-CB[8]-MPyVB-A. **c** $^1$H NMR spectra of digested solutions of U-CB[8]-MPyVB and U-CB[8]-MPyVB-A at 25 °C (DMSO-d$_6$, 500 MHz). **d** Crystal structure of U-CB[8]-MPyVB-A containing two different parts, photoactive motif and photoinert motif. Oxygen atom: red; Carbon atom: gray; Nitrogen atom: blue; Uranium atom: yellow). Photoactive motif (**e**) and photoinert motif (**f**).

decrease only by 94.2% after photodimerization (Supplementary Fig. 13), thus excluding the possibility of photobleaching. As we all know, since the photodimerization of styrene derivatives that breaks molecular conjugation will attenuate the fluorescence signal, the decrease in fluorescence of U-CB[8]-MPyVB-A here offers another evidence of dimerization of the styrene unit in MPyVB ligand. Moreover, the remaining fluorescence intensity may correspond to the MPyVB in Motif 2 of U-CB[8]-MPyVB-A without photodimerization. Moreover, compared with the IR spectrum of U-CB[8]-MPyVB before UV irradiation, the vibration peak at 1622 cm$^{-1}$ that is assigned to C=C absorption band is obviously weakened in U-CB[8]-MPyVB-A after UV irradiation, and the remaining absorption intensity may also belong to MPyVB without photodimerization (Fig. 4c). The shift trend of infrared spectrum in control experiment based on photoactive (HMPyVB$^+$)$_2$@CB[8] complex in aqueous solution is consistent with the above result (Supplementary Fig. 14).

The above results show that the photodimerization of photoactive ligands in these crystalline materials has been successfully realized through structural changes under UV irradiation. With that in mind, a high-quality U-CB[8]-MPyVB-A crystal was selected and subject to UV irradiation in-situ for 1 h, which is subject to single X-ray diffraction analysis to unveil the photo-triggered single-crystal-to-single-crystal (SCSC) transformation. As is shown in Fig. 4d, the photochemistry does occur in Motif 1 (photoactive motif, Fig. 4e) by photodimerization of MPyVB$_D$ and MPyVB$_E$ to give the photodimerization product bisMPyVB with bond distances between C–C bond in cyclobutane being 1.58–1.60 Å. U-CB[8]-MPyVB-A crystallizes in triclinic system with the space group P-1 which is same to U-CB[8]-MPyVB. The asymmetric unit consists of a tetranuclear uranyl center, a MPyVB$_F$, a bisMPyVB ligand, and 1.5 CB[8] (CB[8]$_{D,E}$) molecules. The coordination environment of the tetranuclear uranyl center in U-CB[8]-MPyVB-A is almost identical to that of the tetranuclear uranyl center in U-CB[8]-MPyVB as well as spatial arrangement of the whole structure (Supplementary Fig. 15), which is not described here. While Motif 2 (named photoinert motif, Fig. 4f) does not undergo photoreaction, it shows only slight conformational changes. For instance, the distance between C=C double bonds from two MPyVB ligands changes from 4.50 Å to 4.68 Å. Thermogravimetric analysis shows that U-CB[8]-MPyVB-A has no weight loss before 300 °C, indicating that the compound also has good thermal stability, which is similar to that of U-CB[8]-MPyVB (Supplementary Fig. 3). It is worth mentioning that, when the irradiation time was reduced to as short as 10 min, an elegant intermediate named as U-CB[8]-MPyVB-Int was captured (Supplementary Fig. 16), which contains both possible moieties at the initial site of photoactive motif, a pair of styrene groups without photodimerization and photodimerizing cyclobutene product, respectively. The successful isolation and characterization of U-CB[8]-MPyVB-Int confirms that the photodimerization transformation proceeds through a relatively slow kinetic process.

(37%)/DMSO-d$_6$ after digestion and acidification) in U-CB[8]-MPyVB-A decreases compared with U-CB[8]-MPyVB before UV irradiation, accompanied by the appearance of five groups of new signals (a'-f'), corresponding to the signal of dimeric motif, bisMPyVB (the same phenomenon can be observed in the control experiment, Supplementary Fig. 12). The integration ratio assigned to bisMPyVB is 0.61, which corresponds to about 61% of initial MPyVB in U-CB[8]-MPyVB that underwent photoreactions. In addition, the solid fluorescence experiment was also performed after U-CB[8]-MPyVB subject to UV irradiation (the irradiated U-CB[8]-MPyVB sample is named U-CB[8]-MPyVB-A). As is shown in Fig. 4b, the fluorescence of U-CB[8]-MPyVB-A decreases by 53.3% compared with that before UV irradiation. In the control experiment, the fluorescence intensity of CB[8]-HMPyVB with metal-free (HMPyVB$^+$)$_2$@CB[8] motifs

**Factors for promoting regioselective solid-state photodimerization: supramolecular inclusion and simultaneous coordination**. Because molecular movement in the solid state is greatly restricted, the solid-state photodimerization reaction is more difficult than that in the liquid state. Many factors might have influence on this process, and the distance between C=C double bonds of adjacent photoactive ligands should be one of the most crucial for the realization of solid-state photodimerization[59]. For instance, we have tried to irradiate the crystals of single [HMPyVB]I ligand (Supplementary Fig. 17) with ultraviolet (365 nm) light. Even after a long irradiation time (150 min),

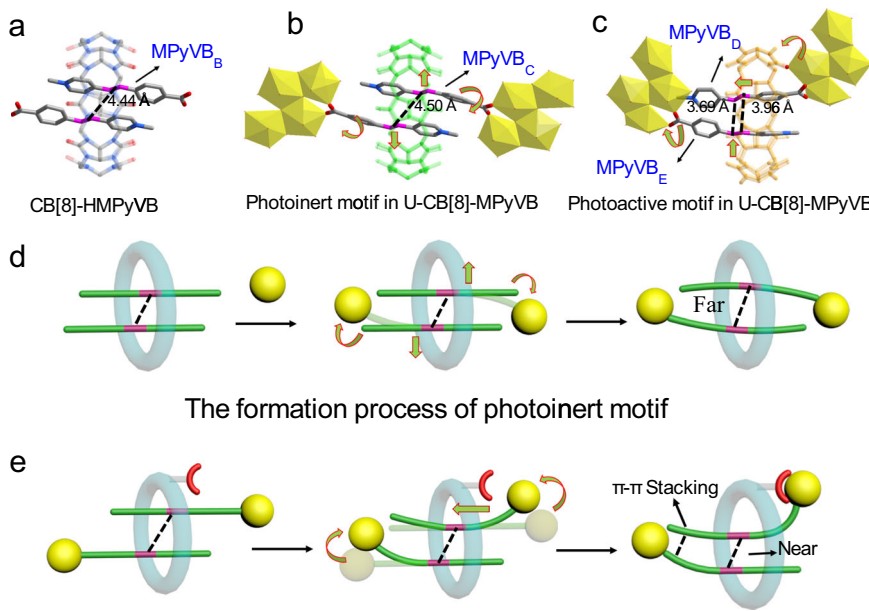

**Fig. 5 Coordination-induced structural changes of (MPyVB)2@CB[8] units in U-CB[8]-MPyVB as compared to CB[8]-HMPyVB and the formation process of photoactive motif and photoinert motif. a** Molecular structure of CB[8]-HMPyVB. **b** Photoinert motif in U-CB[8]-MPyVB. **c** Photoactive motif in U-CB[8]-MPyVB. **d** Cartoon of the formation process of photoinert motif. **e** Cartoon of the formation mechanism of photoactive motif.

the dissolved sample of irradiated [HMPyVB]I crystal has no change in the $^1H$ NMR spectrum (Supplementary Fig. 18). The result is consistent with the single crystal structure of [HMPyVB]I, which shows that the distance between the C=C double bonds in [HMPyVB]I is 5.56 Å, and does not meet the strict condition of [2 + 2] photodimerization.

Furthermore, we introduced CB[8] to the [HMPyVB]I system to obtain CB[8]-HMPyVB structure under hydrothermal conditions. Actually, benefiting from its large hydrophobic cavity, CB[8] has been used as a typical nano reactor to promote some photochemistry reaction because it could incorporate two photoactive guests and limit them to a suitable distance easily by the confinement effect, thus promoting the [2 + 2] photo-dimerization with high efficiency[60,61]. Up to now, CB[8] has been successfully applied to the photodimerization of olefins[62,63], naphthalene[64,65], anthracene[66,67] and coumarin[68,69] in solution, and in these CB[8]-promoting photodimerization processes, high efficiency and high selectivity are always achieved. Here, it is supposed that the confinement of [HMPyVB]I in the cavity of CB[8] might facilitate the photodimerization for CB[8]-HMPyVB. Unfortunately, the distance of 4.44 Å between C=C double bonds in CB[8]-HMPyVB, as mentioned above, is still inadequate to let the [2 + 2] photodimerization reaction happen (Fig. 5a), and $^1H$ NMR monitoring of CB[8]-HMPyVB after light irradiation confirms this point (Supplementary Fig. 19). The results suggest that, although the introduction of CB[8] macrocycles can restrain the photoactive ligands to a certain distance, this current distance is still not enough for the [2 + 2] photodimerization.

Interestingly, the [2 + 2] photodimerization is finally achieved in U-CB[8]-MPyVB, a uranyl-linked metal-organic rotaxane compound. Inspired by the coordination ability of carboxyl groups at both ends of CB[8]-HMPyVB and carbonyl oxygen at the port of CB[8], uranyl ions with flexible coordination modes are further introduced as metal nodes of CB[8]-HMPyVB to construct the coordination compound. It's worth mentioning that there are two kinds of (MPyVB)2@CB[8] motifs in one MORF structure, which can be called, according to the distance of

C=C double bonds and photoresponsive performance, as photoactive motif and photoinert motif, respectively (Fig. 5b, c). Although the chemical components of these two motifs are identical, their molecular conformations as well as photoactivities are totally different due to the differences between them in uranyl coordination patterns. This difference in assembly structure leads to the regioselectivity of the photodimerization reaction in the macrocyclic CB[8] cavity, which will be demonstrated in detail below.

In photoinert motif, after coordination with uranyl ions by the carboxyl group (in bridging bidentate mode) at the end of MPyVB$_C$, the benzoic acid motif will rotate around the C=C bond relative to pyridine ring in order to meet the needs of coordination environment (Supplementary Table 1). The whole MPyVB$_C$ is fixed by the tetranuclear uranyl center through coordination with a downward movement (about 3.9°) with respect to the initial conformation of HMPyVB$_B$ (Fig. 5b and Supplementary Fig. 20). This downward movement of the fixed end will lead to the upward movement of the other end so as to release the possible strain, and finally the distance of C=C bond increase slightly from 4.44 Å to 4.50 Å (the schematic model was proposed in Fig. 5d). The impossibility to carry out [2 + 2] photodimerization at such distance makes this motif to be a photoinert one. On the contrary, the situation is quite different if the photoactive guests and macrocycle parts participate in uranyl coordination at the same time as seen in Fig. 5c. Specifically, uranyl ions in tetranuclear uranyl center first coordinate with the carboxyl group of MPyVB$_D$ through $\mu_2$-($\eta^1$, $\eta^2$) coordination mode. Then in order to further coordinate with the carbonyl oxygen of the CB[8], the whole uranyl-MPyVB$_D$ system will move close to the portal of CB[8], and at the same time, it also pull the carboxyl end to bend up and rotate by 7.9° around C=C bond (Supplementary Fig. 20), thus driving the C=C bond of MPyVB$_D$ to move toward the other photoactive ligand (MPyVB$_E$), and reducing the distance between them. Meanwhile, a proper rotation angle about 15.2° for MPyVB$_E$ helps to stabilize the conformation to a great extent by π-π stacking between these two MPyVB units. Ultimately, simultaneous coordination of

MPyVB and CB[8] with one uranyl center induces the distances between C=C bonds of MPyVB$_D$ and MPyVB$_E$ to reach to a suitable value (3.69 Å and 3.96 Å), which fit well with Schimdt's topochemical criteria (less than 4.2 Å) for [2 + 2] photodimerization in solid state and function as photoactive motifs (the schematic model was proposed in Fig. 5e).

**Structural adaptability and coordination capacity of macrocyclic CB[8].** Generally, the existence of axial guest molecules will greatly increase the steric hindrance around cucurbit[n]uril portals, thus reducing the coordination ability of these rigid macrocycles. On the other hand, we have reported that, self-adaptive CB[6] will be distorted after complexation with the "unsuitable" guest molecules, which is helpful for the carbonyl ports of CB[6] to reduce the steric hindrance from the guest molecules and endows the macrocycle CB[6] with certain coordination ability[70]. In this work, the adaptability of CB[8] also plays a very important role in inducing the formation of photoactive motif. As is shown above, it is the adaptive deformation of CB[8] that makes the wheel macrocyclic molecule participate together with the axle molecule in the coordination with the same uranyl node, and subsequently brings the C=C bonds of encapsulated axle molecules closer. In these interestingly MORF based on CB[8], the axis and the wheel participating in coordination with metal center at the same time. Moreover, this MORF structure even includes both coordinated CB[8] (photoactive motif) and non-coordinated CB[8] (photoinert motif).

In order to quantitatively describe the adaptive deformation of CB[8] in different environments, we define the degree of deformation in adaptive CB[8] at diverse condition by the value $d_{max}/d_{min}$, where the $d_{max}$ and the $d_{min}$ are the maximum distance and minimum distance between two relative carbonyl oxygen in the portals of CB[8] respectively. As is shown in Fig. 6 and Supplementary Table 2, the $d_{max}/d_{min}$ value of CB[8]$_B$ is 1.10, which decreases by 1.79% compared with CB[8]$_A$ ($d_{max}/d_{min}$ = 1.12), indicating that the effect caused by the coordination interaction of guest molecules in the CB[8] cavity is slight. On the contrary, an obvious deformation can be observed compared with the other two macrocycles for the CB[8]$_C$. In fact, the $d_{max}/d_{min}$

value (1.24) of CB[8]$_C$ can prove this deformation, which increases by 10.71% relative to CB[8]$_A$ and is almost six times as great as that of CB[8]$_B$. Moreover, because only one end of carbonyl oxygen in CB[8] participates in the coordination (green atom in CB[8]$_C$, Fig. 6), we wonder whether the effect of metal coordination on the deformation of CB[8] macrocycle is symmetrical. We rotate CB[8] by 180° along the symmetry axis with the plane to get the back view of these CB[8]. The back view of CB[8]$_A$ and CB[8]$_B$ have no change compared with that of front view, which is reasonable because both of two MPyVB participate in the coordination. On the other hand, we can see that the $d_{max}/d_{min}$ values in the front view and the back view of CB[8]$_C$ are quite different because the $d_{max}/d_{min}$ of the back view is 1.14, which is 8.8% lower than the former. The difference between the front view and the back view suggests that uranyl coordination can induce further deformation of CB[8]. In other words, the self-adaptability of CB[8] greatly reduces the steric hindrance of its portals and enables it to participate in the coordination, and in turn, the uranyl coordination of CB[8] portals further exacerbated the deformation of CB[8]. In all, the adaptivity of CB[8] plays a crucial role in the formation of photoactive motif in U-CB[8]-MPyVB through mutual effects of supramolecular inclusion and coordination bonding.

**Photoactuating behavior of bulk crystals of U-CB[8]-MPyVB.** Besides microscopic photo-triggered dynamics as shown above, the photoresponsive behavior of U-CB[8]-MPyVB is first unveiled through its photoinduced motion and macroscopic bending (Supplementary Fig. 10 and Supplementary Movie 1). Besides the rotaxane-based assembly of U-CB[8]-MPyVB to effectively accumulate molecular-scale strain or stress, this macroscopic photoactuating behavior of bulk crystals should be originated from the changes of microscopic structure of U-CB[8]-MPyVB, just like the cases in other photoactuating systems that link macroscopic mechanical response to contraction and expansion of anisotropic lattice[71–74]. Nevertheless, different from other photo-actuated systems, the U-CB[8]-MPyVB system reported here that is a MORF largely relies on both supramolecular inclusion and host-guest coordination. The participation of

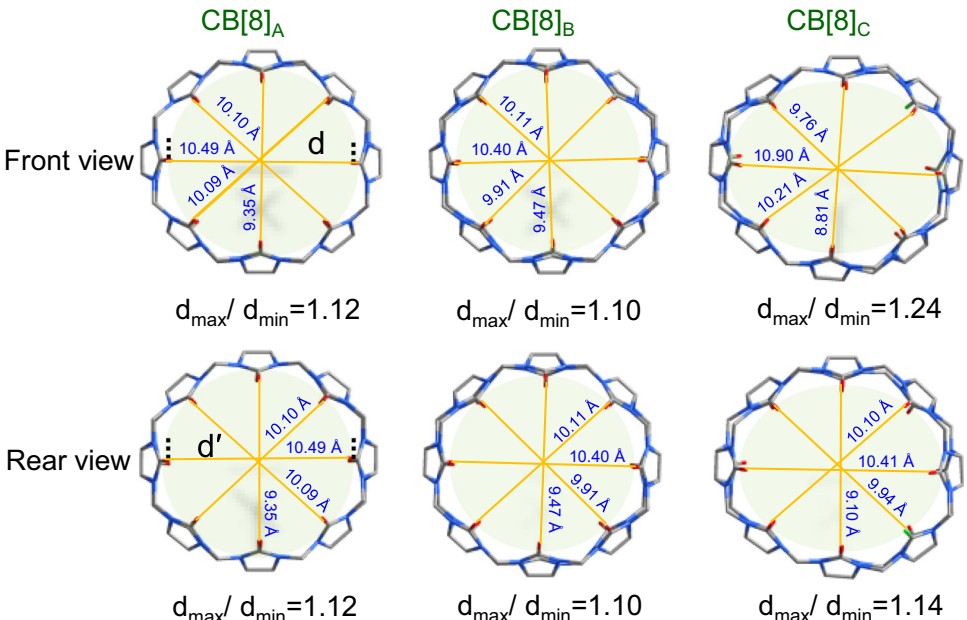

**Fig. 6 Front and rear views of adaptive changes made by CB[8] in different environments.** CB[8]$_A$, from CB[8]-MPyVB; CB[8]$_B$, from photoinert motif of U-CB[8]-MPyVB; CB[8]$_C$, from photoactive motif of U-CB[8]-MPyVB.

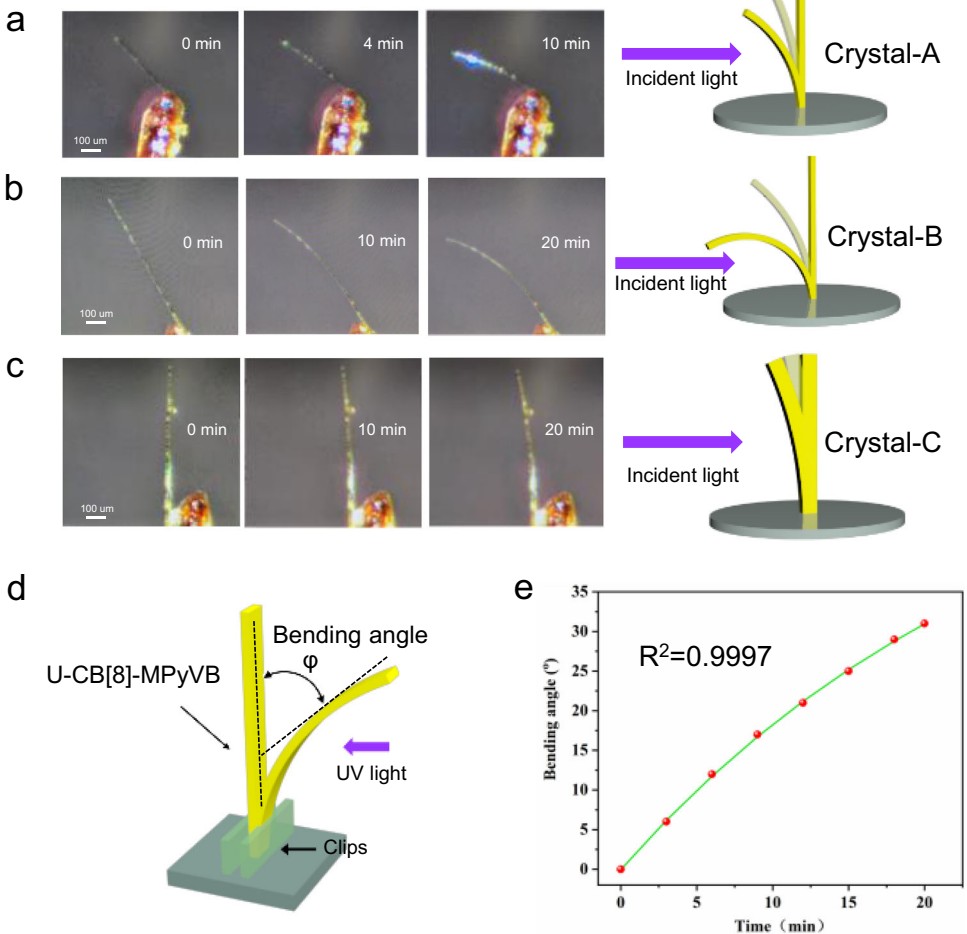

**Fig. 7 The photoinduced bending behavior of bulk crystals of U-CB[8]-MPyVB in different sizes under varying UV irradiation time at 25 °C in air.**
Three kinds of crystals selected for study: Crystal-A (**a**), Crystal-B (**b**), and Crystal-C (**c**). **d** Schematic diagram of experimental design. **e** Function fitting
result for quantifying photoinduced bending behavior of Crystal-B.

rotaxane functional unit in the photoactuating process will make
its macroscopic light actuation behavior be significantly different
from traditional photoactuators. Therefore, continuing efforts are
focused on the characterization of photoactuating behavior of
bulk crystals of U-CB[8]-MPyVB.

Three rodlike crystals (Crystal-A, Crystal-B and Crystal-C) in
different sizes were first selected to figure out the effect of crystal
dimension on photoinduced bending behavior. We fixed one end
of the crystal on the test platform in air so that it stands vertically
and then subject to incident light in specific direction (365 nm,
6 W). As is shown in Fig. 7a, Crystal-A with a length of about
320 μm bends slowly from a straight line towards the direction of
incident light when exposed to ultraviolet light for 10 min. For
Crystal-B with longer size (about 720 μm), as we expected, the
bending degree is higher than that of Crystal-A under the same
illumination time, and the bending phenomenon continues until
20 min (Fig. 7b). Crystal-C with thicker trunk gives a slight
bending movement about 9° within 20 min and the degree of
bending was significantly lower than that of Crystal-A and
Crystal-B (Fig. 7c), proving that a crystal with a larger size
requires larger the cumulative stress in bending progress. The
bending phenomenon of these three pieces of crystals can be seen
more clearly in the Supplementary Movies 2–4. The crystal still
maintains the bending morphology when the incident light
source is removed. Moreover, none of the three crystals show
light-induced damage after a long period of radiation, which
indicates that U-CB[8]-MPyVB has a good stability in air.

These results suggest that this delicate system can convert
photonic energy into mechanical energy at the macroscopic scale,
and the observed light-driven bending is the result of the delicate
balance of crystal size, tensile stress and exposure time to the
incident light source, etc.

In order to further explore the photo-induced bending
behavior of U-CB[8]-MPyVB, we choose Crystal-B as the
representative and quantify the function of bending angle (φ)
changing with time (t) by an exponential function (Fig. 7d). The
dynamic bending process of Crystal-B in 20 min can be expressed
by the Eq. (1):[75]

$$\varphi = k(1 - \exp(-t/\Phi)) \qquad (1)$$

Where k represents the response time constant, t represents the
irradiation time (min) and Φ the actuation response time
coefficient. As is shown in Fig. 7e, the experimental results are
perfectly agreed with the theory, and $R^2 = 0.9997$ can also proves
its correctness, which means the relationship between irradiation
time and bending angle has been established.

For a deeper understanding of how this photomechanical
process of rodlike U-CB[8]-MPyVB crystal under light stimula-
tion and establish the direct relationship between microscopic
photodimerization and macroscopic mechanical bending, detailed
analyses on single-crystal data and crystal face index of U-CB[8]-
MPyVB and U-CB[8]-MPyVB-A are conducted (Fig. 8a). SCSC
transformation from U-CB[8]-MPyVB to U-CB[8]-MPyVB-A
depicted above indicates that the occurrence of the photochemical

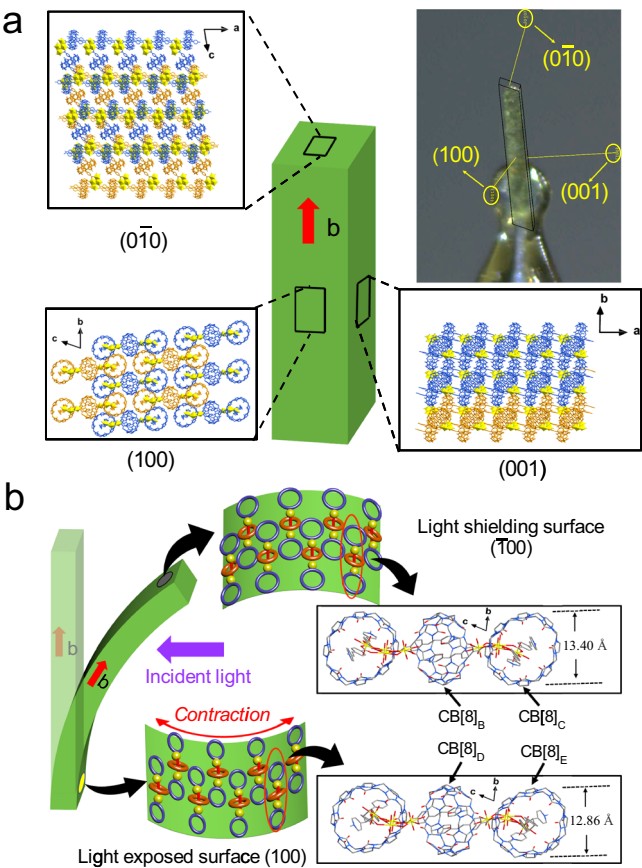

**Fig. 8 A proposed mechanism for the photochemically-induced bending of bulk crystals of U-CB[8]-MPyVB. a** Crystal face index analyses of U-CB[8]-MPyVB. **b** Changes of light exposed surface, (100) plane, and light shielding surface, ($\bar{1}$00$\bar{1}$00) plane, of U-CB[8]-MPyVB crystal when subject to UV light.

reaction in photoactive motif leads to anisotropic contraction/expansion of the whole lattice under ultraviolet light. The unit cell volume of U-CB[8]-MPyVB-A is reduced by 1.66% compared with that of U-CB[8]-MPyVB (from 8823.8 Å$^3$ to 8677.2 Å$^3$, Supplementary Table 3). It can be seen that in the photodimerization process for two MPyVB ligands, the intermediate C=C bonds are close to each other and slowly dimerize into a cyclobutane with the final distance of about 1.60 Å. Meanwhile, methylpyridine and benzoic acid moiety at two ends of bisMPyVB are far away from each other, like a butterfly spreads its wings, from the original 3.47 Å increased to 5.21 Å (Supplementary Fig. 21). As the host, the adaptive CB[8] also takes the corresponding conformational adjustment to match with the change of the guest molecules in the cavity. The vertical distance of CB[8] along the b-axis is contracted from 13.40 Å to 12.86 Å, which is account for the contraction/expansion of unit cell in U-CB[8]-MPyVB (Fig. 8b). Exactly, the length of crystallographic axis b decreased by 2.55% (from 21.21 Å to 20.67 Å, see Supplementary Table 3 for details). While there is no obvious change in a-axis and c-axis because of the existence of strong coordination bonds. Crystal face index analyses reveal that the whole 1-D chains stack layer by layer in (100) plane through weak interactions, which provides the possibility of accumulation of shrinkage stress in b-axis direction. When the light exposed surface (100) plane of U-CB[8]-MPyVB crystal is subject to UV radiation, the crystal plane pointing towards the incident light shrinks along the b axis, while the light shielding surface ($\bar{1}$00$\bar{1}$00) plane keeps it as it is. A similar case of photomechanical bending

of bulk crystals through an anisotropic photoresponsive mechanism was reported by Vittal's group[76]. The single crystal structure of intermediate state (U-CB[8]-MPyVB-Int) as discussed above can also prove the possible coexistence of both reacted and unreacted parts (Supplementary Fig. 16). Finally, the gradual stress accumulation between them accounts for the dynamic bending of the crystal at macroscopic level (Fig. 8b), and the conversion of microscopic structural changes to macroscopic deformation is realized.

As far as we know, the light-response rate of most reported photoresponsive crystal materials based on solid-state photochemical reaction is often very fast (Supplementary Table 4)[48–50,74]. But the photo-induced bending rate of crystal materials reported here is much slower, and falls within the range from hundreds of seconds to thousands of seconds that is two orders of magnitude slower compared with other photoresponsive crystal materials. The main reasons for this difference in photoresponsive dynamics are proposed as followed. (1) As revealed by single-crystal structure, all the MPyVB motifs are encapsulated by CB[8] macrocycles in U-CB[8]-MPyVB. The presence of the inclusion structure may affect the light penetration efficiency and thus reduce the absorption of light by the photoactive groups. (2) Since the diversity of uranyl coordination modes of CB[8], there are both photoactive motif and photoinert motif in the structure of U-CB[8]-MPyVB, which is different from other photoactive crystalline compounds. However, only the photoactive motif in U-CB[8]-MPyVB can undergo photodimerization, while the photoinert motif is just held in place to stabilize the whole structure. In another word, only part of the MPyVB units function after light stimulation, and thus the photochemical transformation efficiency for the U-CB[8]-MPyVB photoactuator system is largely lower than other photoactuator systems. (3) The most important point is that, although the structures of most photoactive ligands will change greatly after photodimerization, the confinement effect of CB[8] in U-CB[8]-MPyVB here restrain the photodimerization-induced structural changes (Fig. 9). Meanwhile, due to the pillaring effect of macrocyclic CB[8] in 3D lattice through intensive hydrogen bonding interactions, the macroscopic bending behavior is mainly originated from the stress related with self-adaptive deformation of CB[8] accumulated in space after the photodimerization of MPyVB in CB[8] cavity. Since the structure change of the macrocycle is only an adaptive adjustment after the photodimerization of the photoacitve guests, the overall structure change is much smaller than that of the guests. Therefore, the photoresponsive effect of U-CB[8]-MPyVB is not that remarkable like those of other photodimerization systems without macrocyclic restriction (see Fig. 9)[50,77]. For instance, after photodimerization, the volume of the whole cell decreases only by 1.66%, and the b-axis, which is the most obvious change in the coordinate axis, also decreases only by 2.55%, which is far lower than the changes of cell parameters of other photoresponsive crystals before and after UV irradiation (Supplementary Table 4). In all, the photoresponsive performance and bending rate of photoactuators are influenced by many factors such as size of the crystal, the intensity of incident light and temperature, etc. It seems that, unlike most photoresponsive crystals are often bent at a large angle in a very short time, U-CB[8]-MPyVB is not sensitive enough to light stimulus here, thus providing possibility to accurately induce the bending motion by light in a controllable manner.

## Discussion

In this work, we successfully synthesized a photoresponsive metal-organic rotaxane framework (MORF) compound, U-CB[8]-MPyVB, from CB[8]-based pseudorotaxane linkers with styrene-derived

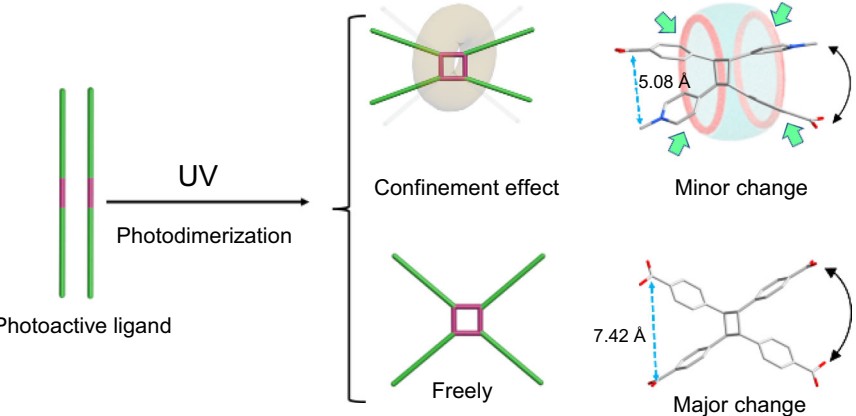

**Fig. 9 Effect of the supramolecular confinement of CB[8] on the ligand conformations after photodimerization.** The confinement effect of CB[8] in U-CB[8]-MPyVB restrain the photodimerization-induced structural changes of photodimerization product, so the distance between methylpyridine and benzoic acid moiety at two ends of bisMPyVB is smaller than that in the traditional photodimerization product.

photoactive MPyVB guest molecules. Benefiting from simultaneous coordination of CB[8] host and MPyVB guest as well as the structural adaptivity of macrocyclic CB[8], one $(MPyVB)_2@CB[8]$ pseudorotaxane motifs turns out to be a photoactive motif, while the other containing non-coordinated CB[8] is still photoinert. Upon irradiation with 365 nm UV light, this uranyl-based MORF U-CB[8]-MPyVB undergoes a single-crystal-to-single-crystal (SCSC) transformation through a regioselective solid-state [2 + 2] photo-dimerization reaction. This photochemical dynamics at molecule scale can further result in macroscopic bending of bulk crystals of U-CB[8]-MPyVB, which indicates high-efficiency assembly of componental units in the lattice of U-CB[8]-MPyVB so as to effectively accumulate molecular-scale strain or stress to macroscopic motion. It is worth mention that, due to the features of MORF system such as the confinement effect of CB[8] and the pillaring role for 3D lattice stacking, the bending rate of this crystal material is two orders of magnitude slower compared with other photoresponsive crystal materials, which provides an opportunity to precisely control bending motion of crystalline state U-CB[8]-MPyVB. We believe these results pave a way to the fabrication of molecular machines with both microscopic and macroscopic dynamics and stimuli-responsive behaviors, especially for reliable photoactuator devices that are capable of delivering precisely controlled mechanical response. The development of such devices will have implications in emerging fields such as optomechanical microdevices and smart microrobotics.

## Methods

### Materials and characterization methods
All chemical reagents are purchased commercially and used directly without further purification. $UO_2(NO_3)_2 \cdot 6H_2O$ (12.55 g, 0.025 mol) was dissolved in 50 mL deionized water to obtain 0.5 M uranyl nitrate mother liquor. The crystal data of [HMPyVB]I, CB[8]-HMPyVB, U-CB[8]-MPyVB, U-CB[8]-MPyVB-Int and U-CB[8]-MPyVB-A are given in Supplementary Table 3. The experimental data results of powder X-ray diffraction (P-XRD) were collected on the Bruker D8 Advance powder-X diffractometer. The copper target provided Kα rays (λ = 1.5406 Å) with a step length of 0.02°. Thermogravimetric analysis (TGA) uses TA Q500 analyzer, and the recorded temperature range is 25–800 °C, the heating rate is 5 °C/min. The infrared (IR) data comes from the Bruker Tensor 27 infrared spectrometer, and the measurement range is 4000–400 cm$^{-1}$. The $^1$H NMR spectrum was recorded by the Bruker AVANCE-III nuclear magnetic resonance spectrometer (500 MHz). ESI-MS spectra were obtained with a Bruker AmaZon SL ion trap mass spectrometer (Bruker, USA). Fluorescence spectra were measured on a Hitachi F-4600 fluorescence spectrophotometer.

### Synthesis of U-CB[8]-MPyVB
Method 1. [HMPyVB]I (18.3 mg, 0.05 mmol), CB[8] (33.2 mg, 0.025 mmol), were added into a 15 mL polytetra-fluoroethylene hydrothermal reactor, and then 2 mL deionized water was added.

The solvent was evenly distributed by ultrasonic vibration for 5 min, and then 140 μL uranyl nitrate solution (0.5 M) was added. After heating at 150 °C for 48 h and natural cooling to room temperature, light yellow rodlike single crystals was obtained, which was suitable for single crystal X-ray diffraction. The crystals were collected by centrifugation, then washed with deionized water (10 mL) for three times, filtered and dried in vacuum. Finally, 35.5 mg light yellow single crystals were obtained with the yield of 49.7%. Method 2. CB[8]-HMPyVB (51.6 mg, 0.025 mmol) was added into a 15 mL polytetrafluoroethylene hydrothermal reactor, and then 2 mL deionized water and uranyl nitrate solution (140 μL, 0.5 M) were added. The solvent was evenly distributed by ultrasonic vibration for 5 min. After heating at 150 °C for 24 h and natural cooling to room temperature, light yellow rodlike single crystals was obtained, which was suitable for single crystal X-ray diffraction. The crystals were collected by centrifugation, then washed with deionized water (10 mL) for three times, filtered and dried in vacuum. Finally, 38.5 mg light yellow single crystals were obtained with the yield of 53.9%.

### Syntheses of U-CB[8]-MPyVB-Int and U-CB[8]-MPyVB-A
Bulk crystals of U-CB[8]-MPyVB were subject to UV irradiation for different time intervals and monitored by single-crystal diffraction determination. U-CB[8]-MPyVB-Int and U-CB[8]-MPyVB-A could be obtained after irradiation for 10 min and 1 h, respectively. For other characterization of U-CB[8]-MPyVB-A, the pristine U-CB[8]-MPyVB (50.0 mg) was grounded well in a mortar and spread flat on the bottom of the beaker. The sample was subject to UV irradiation (365 nm, power of 6 W, the vertical distance between sample and lamp was 5 cm in a dark environment) for an extended period as long as two hours. The yellow powder obtained after the above treatment was collected and used for subsequent characterization.

### Single crystal X-ray-diffraction
The single crystal structure of [HMPyVB]I, CB[8]-HMPyVB, U-CB[8]-MPyVB and U-CB[8]-MPyVB-A were measured on the Bruker D8 VENTURE X-ray diffractometer using a Mo source (λ = 0.71073 Å) at 170 K. All the crystal structures were solved by means of direct methods and refined with full-matrix least-squares on SHELXL-97[78], and refined with full-matrix least-squares on SHELXL-2014[78,79]. Crystallographic data for [HMPyVB]I, CB[8]-HMPyVB, U-CB[8]-MPyVB, U-CB[8]-MPyVB-Int and U-CB[8]-MPyVB-A have been deposited with Cambridge Crystallographic Data Centre, and the CCDC numbers are 2090725-2090727, 2129085 and 2090729.

### Determination the proportion of bisMPyVB in U-CB[8]-MPyVB-A
2.0 mg samples were collected digested with 20 μL concentrated hydrochloric acid (37%) and diluted by DMSO-$d_6$ (0.5 mL). The resulting solution containing MPyVB and bisMPyVB was subjected shortly to $^1$H NMR spectroscopy and the proportion of bisMPyVB content defined as $I_{bisMPyVB}/(I_{MPyVB} + I_{bisMPyVB})$.

## Data availability

All data needed to support the conclusions of this manuscript are provided in the main text or Supplementary Information file or available from the corresponding author upon request.

CCDC 2090725-2090727, 2129085 and 2090729 contains the supplementary crystallographic data for this paper. These data can be obtained free of charge via www.ccdc.cam.ac.uk/data_request/cif, or by emailing data_request@ccdc.cam.ac.uk, or by contacting The Cambridge Crystallographic Data Centre, 12 Union Road, Cambridge CB2 1EZ, UK; fax: +44 1223 336033.

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

## Acknowledgements
We thank the support from the National Science Fund for Distinguished Young Scholars (21925603) and the National Natural Science Foundation of China (22122609, 22076186 and 22076187). The Youth Innovation Promotion Association of CAS (2020014) are also acknowledged.

## Author contributions
W.Q.S. and L.M. conceived the idea and wrote the manuscript. J.S.G. and Y.Y.L. initiated and conducted the photoactive single crystal synthesis and photoresponsive behavior studies, and assistance from L.Y.Y. and J.P.Y. K.Q.H. conducted the single-crystal studies. W.Q.S., L.H.Y., W.F. and Z.F.C. reviewed and edited the manuscript. All authors discussed the results and commented on the manuscript.

## Competing interests
The authors declare no competing interests.
