## [Peer Review File · Nature Communications]

Controllable Photomechanical Bending of Metal-organic Rotaxane Crystal Facilitated by Regioselective Confined-Space PhotodimerizationREVIEWER COMMENTS

Reviewer #1 (Remarks to the Author):

A UO₂ based MOF (or more precisely a MORF) was built using a pseudorotaxane linker comprising two carboxylate axles threaded into a CB8 host. Each identical axle contains a vinyl group and a terminal pyridinium that facilitates the pseudorotaxane formation. The orientation of the vinyl group inside the CB is such that they are amenable to photodimerization. This is shown structurally in Scheme 1b. The idea is that photodimerization would produce a conformational change of the linker which would in turn manifest itself as a (physical) macroscopic change to the material. Although this type of photodimerization is well studied, the authors argue that "the anisotropic assembly and packing of crystalline MORF materials" might be useful for dissipating the resulting stress to the framework and that "incorporating the photoactive axles as linking struts into MORFs to promote greater photo-induced structural changes and internal stress could be a feasible architectural design strategy". In Scheme 1a, they illustrate how this differs from the dynamics of rotation and translation previously reported for MIMs in MOFs. The MORF is interesting and the changes in structures and conformation upon photoirradiation are very cool. I would like to recommend publication but there are some omissions and concerns that need to be addressed.

1) I think this is a very good idea, but the authors need to do a better job of describing what has already been accomplished to put their contributions into perspective. It should be made very clear what has been accomplished in terms of molecular machines in solution and how this differs from the very little that has been demonstrated in the solid state. The way the introductory sentences are written, the first 17 references should refer to solution work. As such, they should acknowledge the full array of molecular machines, robots and assemblers now known, most importantly (but not restricted to) works by Feringa and Leigh as examples.

2) The second paragraph refers to solid state systems, of which there are only a few but does not cite many of the best examples and cites old review materials (e.g., ref. 19 from 2005). Particularly egregious are the omission of the first ever demonstration of a dynamic MIM in a solid-state material (Nat. Chem., 2012, 4, 456) and the citation of a news article (ref. 14) reporting on a molecular shuttle in a MOF rather than the actual article (Nat. Chem. 2015, 7, 514). Here are some newer reports of MIMs and motors in MOFs to go along with ref. 18, (i) Chem. Commun. 2021, 57, 8210; (ii) Chem. Sci. 2021, 12, 394; (iii) Chem. 2021, 7, 202; (iv) Nano. Res. 2021, 14, 417; (v) Faraday Discuss. 2021, 225, 358; (vi) J. Am. Chem. Soc. 2020, 142, 9048; (vii) Cryst. Growth Des. 2019, 19, 5679; (viii) J. Am. Chem. Soc. 2015, 137, 9643. Here are some newer reviews to augment ref. 18 and 19, (i) Chem, 2020, 6, 1604; (ii) Nat. Rev. Chem. 2020, 550; (iii) Trends in Chemistry 2019, 1, 588; (iv) Molecular Machines and Motors: Topics in Current Chemistry, Springer-Verlag, Berlin Heidelberg, 2014, 354: 213; (v) Chem. Soc. Rev. 2012, 41, 5896-5906.

3) Although photodimerization of a pseudorotaxane linker, as described herein, is new, the flexibility of a MIM linker to produce a macroscopic change in a material is not. Please acknowledge and cite the recent work by Sato Nature 2021, 598.

4) Fig. 1 captions. The names of the MOF are not identical but should be – mixes of bold and upper/lower case letters.

5) Fig 1b and 1c show the two different motifs for the linker, but where are they in the extended structure?

6) In the mechanistic study, the formation of a ppt when the CB is not present suggests this order of addition simply prevents pseudorotaxane formation due to the insoluble nature of the resulting UO₂ complex. Presumably there are known UO₂ complexes with carboxylate linkers for comparison. Are these soluble? Compare structures? Reference.

7) The observed motion (not the bending) of the crystal upon irradiation could be due to thermal

effects. This should be noted, discussed, or eliminated as a possibility.

8) Please do not use the phrase <As we all know>.

9) The diagrams infer that there are two different pseudorotaxane dimers in the solid and that one of these dimers can, and does, photodimerize. How can the ¹H NMR of the product show that 61% was dimerized when the max would be 50%?

10) Having X-ray structures of both the original MOF and the dimerized product is beautiful, congratulations. Does the crystal structure of the photo treated product show any disorder at the new site that would infer not all of the sites have reacted?

11) The crystal bending is certainly interesting and a first for these types of dynamic interpenetrated/interlocked solid-state systems. The minimal conformational change of the dimer inside the CB seems like a reasonable explanation for the limited macroscopic changes in the unit cell, but how does this work anisotropically to induce bending? – i.e., do not the two faces of the crystal have to react differently to the light to result in a bend rather than a uniform expansion or contraction of the crystal in one or more directions?

Reviewer #2 (Remarks to the Author):

The manuscript by Shi, Feng and co-workers describes a chemically complex system in which pairs of styrene related molecules, each of which is coordinated to a uranyl-based aggregate, are located inside cucurbit[8]uril units. Upon irradiation with UV light, photodimerization of certain pairs of olefins (but not all) occurs with retention of single crystal character. The crystals also show a mechanical response to the irradiation light which has been quantified. The experimental work appears to be of a high standard and the authors are to be commended on the interpretation of the experimental results. I was impressed by the explanation and proposed mechanism provided for the bending of the crystals in terms of the contraction of the b axis associated with the photodimerization process.

Although there are some minor problems with grammar and expression, the paper is well written with clear explanations of the complex structure supported by excellent schematic diagrams. The manuscript was a pleasure to read and I think the work will be of interest to a wide range of chemists and material scientists. I am pleased to recommend publication of this fine manuscript.

One minor criticism:

I found the labelling in Figure S2 a little confusing. I think it would be clearer if 'Experimental' was labelled in red text to match the red experimental powder pattern and the 'Simulated' was labelled with black text to match the black simulated powder pattern.

Reviewer #3 (Remarks to the Author):

In the Communication, the authors reported a metal-organic rotaxane compound with controllable macroscopic mechanical responses. The regioselective solid-state [2+2] photodimerization and light-triggered single-crystal-to-single-crystal transformation are suggested to induce the macroscopic photomechanical bending of the bulk crystals. Interestingly, the precise control of bending deformation could be realized by adjusting the irradiation time. It is interesting, and the publication is recommended. Other comments:

1) From Figure 3b, the authors suggested that parts of molecules undergo dimerization, however, ¹H NMR is not given before this part of discussion.

2) Line 136, the comparison of the fluorescence intensity is meaningless for the samples in solid-state. The fluorescence quantum yields should be provided.

3) Line 86, the carboxylic acid is shown in Scheme 1B, but it suggested that deprotonation take place.

4) Line 161, what's meaning of metal solvent and ligand solvent?

- 5) D2O-d6 is wrong, it should be D2O.
- 6) Line 187, the bending of the crystal cannot suggest the photoreactions.
- 7) The irradiation time should be provided for the irradiated samples in fluorescence and NMR spectra measurements.
- 8) Lines 410-411, it is mentioned that SCSC is due to a chemical reaction, which causes the changes of crystal lattice. However, the lattice changes in the irradiated part, while the lattice doesn't change in the unirradiated part. It is not single crystal, and it is the mixed crystal.

Point-to-point Response to Reviewers' Comments

Reviewer #1:

A UO_2 based MOF (or more precisely a MORF) was built using a pseudorotaxane linker comprising two carboxylate axles threaded into a CB8 host. Each identical axle contains a vinyl group and a terminal pyridinium that facilitates the pseudorotaxane formation. The orientation of the vinyl group inside the CB is such that they are amenable to photodimerization. This is shown structurally in Scheme 1b. The idea is that photodimerization would produce a conformational change of the linker which would in turn manifest itself as a (physical) macroscopic change to the material. Although this type of photodimerization is well studied, the authors argue that "the anisotropic assembly and packing of crystalline MORF materials" might be useful for dissipating the resulting stress to the framework and that "incorporating the photoactive axles as linking struts into MORFs to promote greater photo-induced structural changes and internal stress could be a feasible architectural design strategy".

In Scheme 1a, they illustrate how this differs from the dynamics of rotation and translation previously reported for MIMs in MOFs. The MORF is interesting and the changes in structures and conformation upon photoirradiation are very cool. I would like to recommend publication but there are some omissions and concerns that need to be addressed.

Response: Thank you for the positive evaluation, and we have done our best to improve our manuscript during revision to meet publication requirements.

1. I think this is a very good idea, but the authors need to do a better job of describing what has already been accomplished to put their contributions into perspective. It should be made very clear what has been accomplished in terms of molecular machines in solution and how this differs from the very little that has been demonstrated in the solid state. The way the introductory sentences are written, the first 17 references should refer to solution work. As such, they should acknowledge the full array of molecular machines, robots and assemblers now known, most importantly (but not restricted to) works by Feringa and Leigh as examples.

Response: Very insightful suggestion. It does help to highlight the difference of our work by describing previous accomplishments and making a comparison of our contributions with them. According to the referee's advice, we have revised the first paragraph that deals with the major accomplishment of molecular machine systems designed to work in solution. Different types of

molecular machines that are molecular switches, molecular pump, molecular motors, molecular muscle, and molecular robot, especially the accomplishment from Feringa and Leigh in solution, are described and important relevant references have been cited. Furthermore, the differences of dynamic performance of molecular machines in the solid state and liquid state have also been elaborated on. The supplemented texts are given as followed. Please see the revised manuscript for more details.

“To date, most molecular machine systems are designed to work in solution where each work unit is isolated from each other and functions independently and incoherently. For example, a variety of molecular machines reported so far including molecular switches, molecular pump, molecular motors, molecular muscle, and molecular robot are capable of making increasingly complex operations in aqueous or nonaqueous environments, where different sets of supramolecular motifs are dispersed and separated individually in solution. However, when extended to a solid-state system with significantly shortened intermolecular distances and massive intermolecular interactions totally different from a solution, it is still challenging to construct MIM-based molecular assemblies with structure dynamics in solid. It is supposed that the dynamic performance of MIMs in such a condensed state could be largely inhibited by great steric hindrance of neighboring atoms and a large number of weak interactions between them.”

2. The second paragraph refers to solid state systems, of which there are only a few but does not cite many of the best examples and cites old review materials (e.g., ref. 19 from 2005). Particularly egregious are the omission of the first ever demonstration of a dynamic MIM in a solid-state material (Nat. Chem., 2012, 4, 456) and the citation of a news article (ref. 14) reporting on a molecular shuttle in a MOF rather than the actual article (Nat. Chem. 2015, 7, 514). Here are some newer reports of MIMs and motors in MOFs to go along with ref. 18, (i) Chem. Commun. 2021, 57, 8210; (ii) Chem. Sci. 2021, 12, 394; (iii) Chem. 2021, 7, 202; (iv) Nano. Res. 2021, 14, 417; (v) Faraday Discuss. 2021, 225, 358; (vi) J. Am. Chem. Soc. 2020, 142, 9048; (vii) Cryst. Growth Des. 2019, 19, 5679; (viii) J. Am. Chem. Soc. 2015, 137, 9643. Here are some newer reviews to augment ref. 18 and 19, (i) Chem, 2020, 6, 1604; (ii) Nat. Rev. Chem. 2020, 550; (iii) Trends in Chemistry 2019, 1, 588; (iv)

Molecular Machines and Motors: Topics in Current Chemistry, Springer-Verlag, Berlin Heidelberg, 2014, 354: 213; (v) Chem. Soc. Rev. 2012, 41, 5896-5906.

Response: We are sorry for our negligence. During revision, a more complete survey of reference literature is afforded. The citation of first report demonstrating a dynamic MIM in a solid-state

material (*Nat. Chem.*, 2012, 4, 456) is supplemented, and the misuse of a news article reporting on a molecular shuttle is corrected by replacing it with the actual article (*Nat. Chem.* 2015, 7, 514). Meanwhile, all the reports recommended by the referee about MIMs and motors in MOFs and important reviews are also cited in the second paragraph when describing the integration of dynamic MIM components with porous metal-organic frameworks (MOFs) and the resultant metal-organic rotaxane framework (MORF) materials.

3. *Although photodimerization of a pseudorotaxane linker, as described herein, is new, the flexibility of a MIM linker to produce a macroscopic change in a material is not. Please acknowledge and cite the recent work by Sato Nature 2021, 598.*

Response: Thanks for the suggestion. The recent work published on *Nature* by Sato's group has been cited in the second paragraph.

4. *Fig. 1 captions. The names of the MOF are not identical but should be – mixes of bold and upper/lower case letters.*

Response: The mistake has been corrected, and similar typos in other parts of the manuscript as well as in supporting information have been also carefully checked and revised.

5. *Fig 1b and 1c show the two different motifs for the linker, but where are they in the extended structure?*

Response: We have redrawn the diagrams in Figure 1 so as to clearly depict the structure of U-CB[8]-MPyVB. Both motifs, **Motif 1** (Figure 1b) and **Motif 2** (Figure 1c), in the extended ladder-like chain are distinguished from each other through marking the CB8 macrocycle in each motif with different colors (Figure 1d), where **Motif 1** with orange-colored CB8 serves as the long vertical bar of ladder-like chain, while **Motif 2** with green-colored CB8 is the short cross bar of ladders. Please see the revised manuscript.

6. In the mechanistic study, the formation of a ppt when the CB is not present suggests this order of addition simply prevents pseudorotaxane formation due to the insoluble nature of the resulting UO₂ complex. Presumably there are known UO₂ complexes with carboxylate linkers for comparison. Are these soluble? Compare structures? Reference.

Response: Thanks for this suggestion. There are several known uranyl complexes with similar organic carboxylate linkers that are insoluble in aqueous solution (*Chem. Eur. J.* 2017, 23,

18074-18083; *Inorg. Chem.*, **2019**, 58, 14075-14084; *Eur. J. Inorg. Chem.*, **2021**, 5077-5084). A comparison between these previously-reported insoluble uranyl compounds with the resulting uranyl complex precipitate formed after mixing $\text{UO}_2(\text{NO}_3)_2 \cdot 6\text{H}_2\text{O}$ and [HMPyVB]I in the aqueous solution prior to the addition of CB[8] macrocycles can well illustrate the precipitation effect closely related to the strong coordination between the ligand and the organic ligand. Since the emerging of the precipitation effect between [HMPyVB]I and uranyl, a different feeding order that first mixing $\text{UO}_2(\text{NO}_3)_2 \cdot 6\text{H}_2\text{O}$ and [HMPyVB]I followed by the addition of CB[8] macrocycles will prevent pseudorotaxane formation between [HMPyVB]I and CB[8] and subsequently fail to obtain the targeted compound U-CB[8]-MPyVB. Along this line, we supplement these examples of insoluble uranyl compounds for comparison, when discussing the assembly mechanism of U-CB[8]-MPyVB, with the resulting uranyl complex precipitate formed after mixing $\text{UO}_2(\text{NO}_3)_2 \cdot 6\text{H}_2\text{O}$ and [HMPyVB]I in the aqueous solution.

7. The observed motion (not the bending) of the crystal upon irradiation could be due to thermal effects. This should be noted, discussed, or eliminated as a possibility.

Response: Thanks for this valuable advice. We have carried out a detailed analysis and discussion of the accompanying thermal effects during the irradiation process in order to exclude possible interference caused by thermal effects. First, we monitor temperature changes of mineral oil during the irradiation by a UV lamp with very lower power (365 nm, power of 6 W). As is shown in Figure S11a (see below), the initial value of temperature is about 16 °C (289 K), and the temperature only reached ~30 °C (303 K) after irradiation for 1 h. Based on the minor temperature change tested above, a control experiment by directly heating crystals in mineral oil over this temperature range is conducted, and the experimental results reveal that the crystals keep silent and show no sign of motion, even up to 40 °C. Furthermore, we determine the crystal structure of crystal samples that are immersed in mineral oil or fixed on the loop and subject to heating to 40 °C (Figures S11b-c, see below). The structure of crystals after heating shows no change with that of the pristine sample. Therefore, we exclude the possibility of crystal motion caused by external thermal disturbance, and the crystal motion can only be driven by its own internal irradiation-induced structural changes. The corresponding discussion has been supplemented in the revised manuscript.

Figure S11. Temperature changes induced by photothermal effect and its possible effects on the motion of the crystal under UV lamp (365 nm, power of 6 W) irradiation condition. **(a)** The temperature changes at different irradiation time induced by photothermal effect. **(b-c)** Changes of crystals immersed in mineral oil **(b)** and fixed on the Loop **(c)** when subject to heated atmosphere at 40 °C after 60 minutes.

8. Please do not use the phrase.

Response: The ‘solid phase’ throughout the main text has been changed to ‘solid state’ in the revised manuscript.

9. The diagrams infer that there are two different pseudorotaxane dimers in the solid and that one of these dimers can, and does, photodimerize. How can the ^1H NMR of the product show that 61% was dimerized when the max would be 50%?

Response: The photoactive motif (**Motif 1**) and photoinert motif (**Motif 2**) have been marked in the extended chain structure, in which the proportion of photoactive regions is two-thirds (two photoactive motif versus one photoinert motif) rather than one-half. So theoretically, the maximum yield of photodimerization product is ~66.6% rather than 50%. Therefore, it is reasonable to observe a portion of 61% that has been dimerized.

10. Having X-ray structures of both the original MOF and the dimerized product is beautiful, congratulations. Does the crystal structure of the photo treated product show any disorder at the new site that would infer not all of the sites have reacted ?

Response: Indeed, since the data for single crystal characterization are statistically averaged results

within the entire crystal, it can be inferred that structural disorder is observed in single-crystal structure determination when part of the active sites is unreacted. Moreover, this process should be a kinetic process closely related to the irradiation time. In the crystal structure of U-CB[8]-MPyVB-A that was subject to UV irradiation for a relatively longer period of time (1 h), no structural disorder at the photodimerization site was observed. Meanwhile, in order to capture the intermediate state of dimerized product so as to trace the photodimerization process, we reduced the UV irradiation time (10 min), and collected X-ray diffraction crystal data of the corresponding photo-treated product (namely as U-CB[8]-MPyVB-Int). Single-crystal structure analysis of U-CB[8]-MPyVB-Int (Figure S16, see below) reveals that, the intermediate state of photodimerization, as expected, shows some disorder at the photodimerizable sites. Specifically, both possible moieties, a pair of styrene groups without photodimerization and photodimerizing cyclobutene product, can be found in the same site of photoactive motif in U-CB[8]-MPyVB-Int, which means that not all of the sites have reacted when the irradiation time is not enough. Corresponding description and detailed discussion on this issue as well as the crystal structure of U-CB[8]-MPyVB-Int have been revised and supplemented in the main text and SI during revision.

Figure S16. Single crystal structure of U-CB[8]-MPyVB-Int. (a) Asymmetric unit of U-CB[8]-MPyVB-Int. Crystal structure of photoactive motif of U-CB[8]-MPyVB-Int containing two possible different parts, unreacted part and reacted part. (b-c) Two possible parts of photoactive motif in U-CB[8]-MPyVB-Int: unreacted part (b) and dimerized part (c) (oxygen atom: red; carbon atom: gray; nitrogen atom: blue; uranium atom: yellow).

11. The crystal bending is certainly interesting and a first for these types of dynamic interpenetrated/interlocked solid-state systems. The minimal conformational change of the dimer inside the CB seems like a reasonable explanation for the limited macroscopic changes in the unit cell, but how does this work anisotropically to induce bending? – i.e., do not the two faces of the crystal have to react differently to the light to result in a bend rather than a uniform expansion or contraction of the crystal in one or more directions?

Response: This is true that the two opposite faces (light exposed surface plane and light shielding surface plane) of the photo-responsive crystal sample react differently to the light to result in a bending deformation (a similar mechanism was reported by Vittal's group, *Chem. Mater.* **2021**, 33, 4621-4627). Specifically, the photodimerization reaction inside the CB[8] accounts for the adaptive contraction of CB[8] (from 13.40 Å to 12.86 Å along *b*-axis). The weak interactions between CB[8] along *b*-axis make it possible to accumulate the shrinkage stress, while it is impossible to contract or elongate in *a*-axis and *c*-axis because of the restraint by relatively strong coordination bonds. As a result, the crystal plane pointing towards the incident light shrinks and the light shielding surface plane keeps it as it is. Finally, the gradual stress accumulation between these two planes leads to macroscopic bending of the irradiated crystal. It should be mentioned that the intermediate state structure (U-CB[8]-MPyVB-Int) proves simultaneous existence of possible reacted and unreacted parts in the crystal of photoactive U-CB[8]-MPyVB once there are differences in the light intensity received by different parts of the crystal sample.

Reviewer #2:

*The manuscript by Shi, Feng and co-workers describes a chemically complex system in which pairs of styrene related molecules, each of which is coordinated to a uranyl-based aggregate, are located inside cucurbit[8]uril units. Upon irradiation with UV light, photodimerization of certain pairs of olefins (but not all) occurs with retention of single crystal character. The crystals also show a mechanical response to the irradiation light which has been quantified. The experimental work appears to be of a high standard and the authors are to be commended on the interpretation of the experimental results. I was impressed by the explanation and proposed mechanism provided for the bending of the crystals in terms of the contraction of the *b* axis associated with the photodimerization process.*

Although there are some minor problems with grammar and expression, the paper is well written with clear explanations of the complex structure supported by excellent schematic diagrams. The manuscript was a pleasure to read and I think the work will be of interest to a wide range of chemists and material scientists. I am pleased to recommend publication of this fine manuscript.

Response: Thank you for the positive comments.

One minor criticism:

I found the labelling in Figure S2 a little confusing. I think it would be clearer if 'Experimental' was labelled in red text to match the red experimental powder pattern and the 'Simulated' was labelled with black text to match the black simulated powder pattern.

Response: Thanks for the reviewer's suggestion and Figure S2 has been revised. 'Experimental' has been labelled in red and 'Simulated' has been labelled in black.

Reviewer #3:

In the Communication, the authors reported a metal-organic rotaxane compound with controllable macroscopic mechanical responses. The regioselective solid-state [2+2] photodimerization and light-triggered single-crystal-to-single-crystal transformation are suggested to induce the macroscopic photomechanical bending of the bulk crystals. Interestingly, the precise control of bending deformation could be realized by adjusting the irradiation time. It is interesting, and the publication is recommended. Other comments:

1. From Figure 3b, the authors suggested that parts of molecules undergo dimerization, however, ¹H NMR is not given before this part of discussion.

Response: We have rearranged the parts in Figure 3 by placing ¹H NMR spectra before fluorescence and IR spectra, and the discussion in the main text has been adjusted in the revised manuscript accordingly.

2. Line 136, the comparison of the fluorescence intensity is meaningless for the samples in solid-state. The fluorescence quantum yields should be provided.

Response: Thanks for the valuable suggestion. The fluorescence quantum yields of CB[8]-HMPyVB and U-CB[8]-MPyVB have been provided and related discussion has been added in the revised version as follows:

“Compared to strong fluorescence of supramolecular complex CB[8]-HMPyVB, the fluorescence of U-CB[8]-MPyVB is greatly weakened (the fluorescence quantum yield changes from 2.8% for CB[8]-HMPyVB to ~0% for U-CB[8]-MPyVB after uranyl coordination)”

3. Line 86, the carboxylic acid is shown in Scheme 1B, but it suggested that deprotonation take place.

Response: We are sorry for our negligence and Scheme 1 and Figure 2 have been revised. The deprotonated ligand has been reformulated.

4. Line 161, what's meaning of metal solvent and ligand solvent?

Response: For clarity, the corresponding description has been revised as “UO₂(NO₃)₂ solution and [HMPyVB]I solution (molar ratio between them is 1:1), respectively”.

5. D₂O-d₆ is wrong, it should be D₂O.

Response: We are sorry for our negligence and “D₂O-d₆” has been changed to “D₂O”.

6. Line 187, the bending of the crystal cannot suggest the photoreactions.

Response: Thanks for the reviewer's suggestion, it is true that our description is not rigorous enough. The corresponding description has been revised as follows:

“This interesting photoresponsive phenomenon indicates that the photo-triggered reactions may occur within the crystal of U-CB[8]-MPyVB during incident irradiation. The ¹H NMR experiment reveals that there is photodimerization product, 1,3-bis(4-methylpyridin)-2,4-bis (benzoic acid) cyclobutane (bisMPyVB), in the digested sample of U-CB[8]-MPyVB.”

7. The irradiation time should be provided for the irradiated samples in fluorescence and NMR spectra measurements.

Response: The irradiation time for the irradiated samples was supplemented and described in the “Methods” section as follows:

“Synthesis of U-CB[8]-MPyVB-A: U-CB[8]-HMPyVB (50.0 mg) was grounded in a mortar and spread flat on the bottom of the beaker. The sample was subject to adequate irradiated with a UV lamp (365 nm, power of 6 W, the vertical distance between sample and lamp was 5 cm in a dark environment). After two hours, the yellow powder was collected and used for characterization.”

8. Lines 410-411, it is mentioned that **SCSC** is due to a chemical reaction, which causes the changes of crystal lattice. However, the lattice changes in the irradiated part, while the lattice doesn't change in the unirradiated part. It is not single crystal, and it is the mixed crystal.

Response: We agree that, the long rod-like **U-CB[8]-MPyVB** crystal that shows a macroscopic bending upon UV irradiation have both the reacted and unreacted part at light exposed surface and light shielding surface, respectively, and should be recognized as mixed crystal, but not single crystal. But there is no contradiction with the use of single-crystal-to-single-crystal (**SCSC**) transformation, because the **SCSC** transformation here is referred to the photo-triggered chemical transformation from **U-CB[8]-MPyVB** to **U-CB[8]-MPyVB-A**, the nature of which has been clearly identified by single-crystal X-ray diffraction analyses of both compounds to be a single-crystal-to-single-crystal (**SCSC**) transformation process. Actually, the same term of ‘**SCSC** transformation’ triggered by photodimerization can be referred to several literatures reported previously: *J. Am. Chem. Soc.*, **2021**, 143, 5636-5642; *Nat. Commun.*, **2020**, 11, 2808; *Angew. Chem. Int. Ed.*, **2014**, 53, 2143-2146; *J. Am. Chem. Soc.*, **2020**, 142, 20117-20123; *Chem. Commun.*, **2020**, 56, 1984-1987; *J. Am. Chem. Soc.*, **2020**, 142, 6180-6187; *Angew. Chem. Int. Ed.*, **2019**, 58, 9453-9458; *Angew. Chem. Int. Ed.*, **2014**, 53, 414-419, etc. Herein, we intend to unveil the mechanism of photo-triggered crystal macroscopic bending of long rod-like **U-CB[8]-MPyVB** crystal by analyzing the **SCSC** transformation process from **U-CB[8]-MPyVB** to **U-CB[8]-MPyVB-A** discussed above, which actually involves a contraction of unit cell and whole lattice upon irradiation. For this purpose, detailed analyses on single-crystal data and crystal face index of **U-CB[8]-MPyVB** and **U-CB[8]-MPyVB-A** are conducted and discussed. In order to avoid any misunderstanding, we supplement the qualifier of **SCSC** and modify the description as “...*The SCSC transformation from U-CB[8]-MPyVB to U-CB[8]-MPyVB-A indicates that photochemical reaction in photoactive motif leads to a slight contraction of the whole lattice under ultraviolet light.*”

REVIEWER COMMENTS

Reviewer #1 (Remarks to the Author):

Thank you very much. The authors have done an excellent job of addressing all of my concerns, This is now an significant and well written contribution to the area. I recommend publication of this revised manuscript.

Reviewer #3 (Remarks to the Author):

The authors have revised the manuscript according to the comments from Referees. The publication is recommended.

REVIEWERS' COMMENTS

Reviewer #1 (Remarks to the Author):

Thank you very much. The authors have done an excellent job of addressing all of my concerns, This is now an significant and well written contribution to the area. I recommend publication of this revised manuscript.

Response: Thanks for the positive evaluation.

Reviewer #3 (Remarks to the Author):

The authors have revised the manuscript according to the comments from Referees. The publication is recommended.

Response: Thanks for the positive evaluation.